# Telemedicine in the Management of Parkinson’s Disease: Achievements, Challenges, and Future Perspectives

**DOI:** 10.3390/brainsci12121735

**Published:** 2022-12-19

**Authors:** Esther Cubo, Pedro David Delgado-López

**Affiliations:** 1Neurology Department, Hospital Universitario Burgos, University of Burgos, 09006 Burgos, Spain; 2Neurosurgery Department, Hospital Universitario Burgos, University of Burgos, 09006 Burgos, Spain

**Keywords:** telemedicine, teleneurology, telecare, movement disorders, tele-education, new technology

## Abstract

Background: As the global population grows, there is an increasing demand for neurologic consultation that prompts new ways to reach more patients. Telemedicine can provide an accessible, cost-effective, and high-quality healthcare services. Objectives: In this article, we highlight recent developments, achievements, and challenges regarding outcomes, clinical care, tele-education, teletreatment, teleresearch, and cybersecurity for telemedicine applied to Parkinson´s disease (PD) and other neurological conditions. Results: A growing body of evidence supports the feasibility and effectiveness of telemedicine tools for PD and other movement disorders. Outcome variables regarding satisfaction and efficacy in clinical care and specific issues about education, research, and treatment are reviewed. Additionally, a specific legal framework for teleconsultation has been developed in some centers worldwide. Yet, the implementation of telemedicine is conditioned by the limitations inherent to remote neurological examination, the variable computer usage literacy among patients, and the availability of a reliable internet connection. At present, telemedicine can be considered an additional tool in the clinical management of PD patients. Conclusions: There is an increasing use of remote clinical practice regarding the management of PD and other neurological conditions. Telemedicine is a new and promising tool aimed at special settings and subpopulations.

## 1. Introduction

Neurological disorders are increasingly recognized as major causes of death and disability worldwide [1,2]. According to the World Health Organization (WHO), in collaboration with the World Federation of Neurology, adequate healthcare resources for patients with neurological disorders are lacking in many countries. Inequalities in access to neurological care are particularly dramatic among patients living in low-income countries and underserved regions of the world [3].

Traditional medical practice may not always be the most efficient or convenient way to care for neurological patients. With an increasing demand for neurological services in a fast-growing population, technology can help in providing healthcare to more patients. As remote technologies develop, it is key that practitioners maintain high-quality healthcare standards, at least equivalent to traditional in-person visits. Information and communication technologies (ICTs) can help to address some of the current healthcare challenges, such as accessibility, both in developed and underserved countries, and provide cost-effective and high-quality services [4]. Telemedicine uses ICTs to overcome geographical barriers and it is particularly beneficial for rural and underserved communities [5].

Telemedicine was first used in neurology to facilitate access to specialized treatment for acute stroke. Further, over the last decade, it has been shown to be particularly suitable for subjects with Parkinson’s disease (PD) and other movement disorders. Spurred by the COVID-19 pandemic, telemedicine in its various forms has become a widely considered topic [6]. In March and April 2020, the International Parkinson and Movement Disorder Society Telemedicine Study Group initiated a survey across 40 countries to analyze four domains of telemedicine: legal regulations, reimbursement, clinical usage and tools, and barriers within each region [7]. This survey confirmed a global increase in telemedicine usage, including telephone calls, messaging apps, and video visits. However, although the growth of telemedicine use was relevant in both low- and high-income countries, there was a significant variability regarding regulations and infrastructure supporting telemedicine between low- and high-income countries [7].

Given the growing interest in telemedicine for PD and other movement disorders, in this article, we discuss the rationale of telemedicine, and some specific features, such as (1) the *variability in healthcare delivery* during the pandemic COVID-19 on individuals with PD and other movement disorders; (2) the *outcome variables* for telemedicine; (3) *telecare*, including *tele-expertise* (seeking a second specialized opinion regarding parts of the patient’s medical file), and *teleconsultation* (remote patient consultations); (4) *tele-education* for health professionals and patients/caregivers; (5) *teletreatment solutions* for PD and other movement disorders; (6) *teleresearch*; (7) *cyber security; and* (8) *barriers and limitations for using telemedicine*. According to recent publications, many telemedicine studies deal with videoconferencing technology, especially for patients with PD (Table 1).

## 2. Changes in Care in Response Due to the COVID-19 Pandemic

Studies from different countries have demonstrated that individuals with PD experienced worsening motor and nonmotor symptoms during the pandemic [8,9,10,11,12,13]. Similarly, the pandemic has negatively affected individuals with other movement disorders such as multiple system atrophy, dystonia, and tic disorders [14,15,16]. Overall, telemedicine applied to PD patients increased from 9.7% prior to the pandemic to 63.5% during the pandemic [17]. In fact, we learned from the pandemic that many aspects of healthcare for patients with movement disorders could be provided remotely [8].

## 3. Outcomes for Telemedicine in Movement Disorders

The ongoing COVID-19 pandemic has prompted telemedicine as a novel and effective tool for the management of movement disorder conditions. In a recent review of case series and randomized controlled pilot trials in PD [18], a heterogenous group of outcomes was identified, including feasibility, satisfaction, and efficacy. Regarding feasibility, defined by the percentage of participants who finalized at least one visit [19], overall, telemedicine was considered feasible and efficient in connecting clinicians with patients with different needs [20]. Effective follow-up visits for people with PD were established for institutionalized patients, patients’ homes, healthcare facilities, rural communities, advanced PD patients [20,21,22,23], and other movement disorders, such as functional motor disorders [24].

The other most commonly reported outcome was acceptability or satisfaction. Overall, studies assessing satisfaction found that the majority of patients and clinicians were satisfied with the use of telemedicine [18]. In Huntington’s disease (HD), surveys containing questions about participants’ experience with the predictive testing process found similar satisfaction between the in-person and telemedicine groups [25]. In a recent PD study, in which satisfaction was measured with the Likert rating scale, ranging from “very satisfied” to “very unsatisfied”, high satisfaction rates for telemedicine were found [26]. In contrast, studies published before 2006 reported suboptimal video quality in 82 of 100 visits, most likely attributed to technical problems [27].

Regarding efficacy and validity/reliability, therapeutic interventions using telemedicine, delivering voice treatment and physical therapy for PD, and behavioral interventions in children with tic disorders were considered effective and showed similar results to the in-person visit groups [28,29,30,31]. In fact, PD studies comparing in-person and telemedicine visits showed high intraclass correlation coefficients, although agreement decreased when PD patients with significant motor fluctuations were included [22,32,33].

Other outcomes reported in telemedicine studies for movement disorders include the description of participant experiences, the perception of quality of care, convenience, usability, and comfort [34,35,36,37,38]. According to these outcomes, telemedicine was found convenient regarding the perception of quality of care, including extended access to multidisciplinary care, decreased travel burden, and suitability of in-home visits [6]. However, concerns about the implementation of telemedicine included privacy issues regarding the clinician–patient relationship, diagnostic accuracy due to limited neurological examination, and lack of information on cost-effectiveness [39].

## 4. Telecare for Parkinson’s Disease and Other Movement Disorders

A fascination for telemedicine is based upon its ability to overcome the space barrier that limits access to movement disorder experts [4]. Although the typical outpatient visit to the clinic offers face-to-face contact, it provides, at best, a suboptimal perspective of the patient’s real functioning at home. In fact, videoconferences can facilitate clinicians better evaluation of patients in their own usual environment [4].

Telemedicine is particularly capable of assessing patients with PD and other movement disorders because much of the neurological exam findings are visual. Parts of neurological examinations can be performed during teleconsultations, providing equivalent results to in-person assessment and evaluation of candidacy for advanced PD therapies [40,41]. These teleconsultations included not only synchronous encounters (videoconferences), but also telephone and e-consultations as well (for a step-by-step guide, see the International PD and Movement Disorder Society recommendations) [42].

For other movement disorders, HD teleclinics have also been shown to be feasible, with some modifications in the examination technique. Cognitive screening can be performed using written portions of the Montreal Cognitive Assessment (MOCA) presented with screenshots [43] and genetic counseling for HD is routinely conducted using telemedicine in some centers [44]. In a pilot study of 11 HD patients, Bull et al. were able to perform successive follow-up visits after the first in-person assessment [25]. Patients were examined at home with a web camera. Although there were some technical limitations in assessing ocular movements, balance, and gait, most elements of the physical examination were reliable. Other multicenter, international studies have also shown increased use of telemedicine for HD [45], ranging from discussions with other clinical providers to patient visits or between-visit support.

For movement disorders consultations, the International Parkinson’s Disease and Movement Disorder Society has sponsored several telemedicine programs in underserved areas over the last few years. Asynchronous Consultation in Movement Disorders (ACMD) is a specialized program conducted in Africa [46], which eliminates the difficulties of scheduling virtual clinic visits in different time zones. In the ACMD program, the specialized consultant merely provides advice to the local physician, who has the principal responsibility for the patient. The report provided by the consultant usually incorporates a differential diagnosis, a list of follow-up questions for consideration, an empiric plan of care, and relevant academic literature. In 2018, 12 out of 51 clinical cases (43%) presented using the ACMD platform were related to dystonia, myoclonus, and dyskinesias, and none contained queries regarding PD [47], highlighting the difficulties of diagnosing hyperkinetic movement disorders in such underdeveloped areas.

Videoconference-based telecare has been shown effective for patients and caregivers [48]. At times, patients might need some help during the course of the remote visit. This assistance can be provided by caregivers or other health-related personnel (*telepresenters*) available at telemedicine facilities.

Remote monitoring with the use of accelerometers and mobile applications can be useful for motor assessment in PD [49,50]. Although strong evidence and clinical guidelines still lack, it is believed that remote patient telemonitoring can provide objective data that may supplement the MDS-Unified Parkinson’s Disease Rating Scale (UPDRS), and help to obtain complete outcome measures in clinical research and optimize pharmacological and nonpharmacological treatments [40,51].

Another excellent example of a successful and established multidisciplinary regional care model in PD is the ParkinsonNet model implemented in the Netherlands [52]. In this program, medical and allied healthcare personnel deliver interventions integrated into regional community networks dispersed throughout the country with PD-specific therapists [53]. In this program, the use of specialized occupational therapy delivered in the community setting improved self-perceived daily functioning, a better quality of care, fewer PD-related complications, and lower total healthcare costs compared to usual care [54].

## 5. Tele-Education for Parkinson’s Disease and Other Movement Disorders

Successful integration of telemedicine training among general practitioners and allied health professional programs is vital to expand access to an ever-increasing demand for neurologic consultation. In recent years, tele-education for patients with movement disorders, caregivers, and other health professionals has become popular. Access to specialized care and education remains poor in certain subpopulations, especially among those with limited resources. In this regard, telemedicine and tele-education are particularly needed in remote areas facing a shortage of general practitioners and specialists.

Telemedicine is also an effective strategy for recruiting and retaining physicians in underserved areas. Distance learning can break their professional isolation and reduce the related stress [46]. In this regard, in 2014, a tele-education PD program was conducted in Cameroon, sponsored by the International Parkinson’s Disease and Movement Disorder Society [55]. Twenty lectures were given using synchronous video conferences throughout the year. The events connected movement disorder experts with 33 health professionals (52.4% women), including 16 doctors and 17 allied health professionals. Videoconferences were completed in 80% of the cases (feasibility), and attendees’ participation ranged from 20% to 70% (adherence), with high satisfaction and improvement in medical knowledge (effectiveness).

Another interesting pilot study presented a tele-education program that covered hyperkinetic and hypokinetic movement disorders for medical students [56]. It included 151 undergraduate medical students, 79.4% from Argentina and 20.6% from Cameroon. Feasibility was acceptable with 100% and 85.7% of the videoconferences completed in Argentina and Cameroon, respectively, and medical knowledge improved similarly in both countries. Likewise, the conduction of fellow and neurology resident training programs, locally, nationally, and internationally, was possible with the supervision of virtual clinical visits of patients carrying movement disorders [57,58]. Overall, these programs showed a statistically significant improvement in the learning process and the comfort of patients. Tele-education has also been shown to be feasible for caregivers and patients’ support groups [59].

## 6. Teletreatment for Parkinson’s Disease and Other Movement Disorders

During the COVID-pandemic, DBS parameters for PD could be successfully adjusted remotely [60] during online therapeutic sessions. Efficacy and satisfaction rates with the remote DBS adjustment sessions were comparable to in-clinic DBS adjustments [60]. Telemedicine has also been used in a small, open-label study to assist with levodopa-carbidopa intestinal gel infusion titration [38]. In this study, telemedicine was considered an efficient and accepted tool, technically feasible, and satisfactory to patients, neurologists, and nurses [38]. Likewise, intensive adjustment of the apomorphine dose in response to patients’ motor fluctuations could be conducted either manually by the patient or automatically to detect motor symptoms [61]. Rodriguez et al. report a tendency towards improved symptom control (shorter time in off and fewer rescues needed) during the intervention periods using a semi-automatic control of the infusion pumps by motor sensors, where the dose of apomorphine was changed depending on the patients’ PD motor state [61]. Specifically, telemonitoring technologies, in addition to videoconferencing, seem to be useful in identifying patients who may be candidates for advanced PD therapies [62].

For patients with cervical dystonia, telemedicine was found useful to evaluate the efficacy of botulinum toxin during its peak effect. Overall, the agreement between the Toronto Western Spasmodic Torticollis Rating Scale (TWSTRS) motor severity subscale assessed in clinic vs. telemedicine visits was excellent, with good acceptance among users [63]. Other candidates for teletreatment are those homebound persons with movement disorders and multimorbid conditions. In such cases, multidisciplinary teams providing various nonpharmacological interventions at home, such as physical therapy, speech therapy, psychiatric interventions, and cognitive training, have demonstrated effectiveness, high adherence rates, and significant effects on their daily routine and functioning [6,64,65].

## 7. Teleresearch for Parkinson’s Disease and Other Movement Disorders

New advances in clinical drug development for PD are needed. With the advance of telemedicine, there is a growing opportunity to enroll patients in clinical trials, across large geographical areas in a relatively short period of time. A new generation of digital equipment—including smartphones, wearable sensors, and in-home monitors—permits frequent and objective evaluations of PD, capturing the functional status in real-world settings [50]. Recently, Schneider et al. published a protocol describing the infrastructure needed for virtual follow-ups of clinical trial subjects, including the changes in smartphone-based evaluations, online patient-reported outcomes, remote professional assessments, and the delivery of innovative digital markers of PD disability and progression [66]. In this ongoing project, out of 226 enrolled individuals with PD, 181 (80%) successfully downloaded the study’s smartphone application, and 161 (71%) concluded patient-reported outcomes on the online platform. The results of this study will provide data about the comparison of established clinical endpoints with novel digital biomarkers and thereby inform the longitudinal follow-up of clinical trial participants and the design of future clinical trials.

On the other hand, barriers to participation in trials are numerous and include troublesome financial, travel, and caregiver requirements that may be particularly problematic for individuals with mobility difficulties and cognitive impairment [67]. However, acknowledging the restrictions posed by the pandemic regarding the conduction of clinical trials, telemedicine offers the opportunity to recenter the patient in clinical research and improve the clinical research process [8]. Decentralized studies might change the old-style research structure, moving it from the clinic to the patient’s home. This may include, for example, remote assessment with video visits, collection of real-world data using digital devices, in-home safety evaluations, home delivery of the study drug, and the collection of biological specimens [8]. However, because not all studies are suitable for a fully decentralized approach, hybrid research studies may offer the best of both worlds [8].

Further, the *metaverse* represents a promising internet-based technology that produces an immersive virtual experience of alternative reality for users [68]. This technology opens the door to virtual visits in virtual hospitals, education to patients, medical students, and multidisciplinary care. Yet, the cost of the technology, and some privacy and credibility issues, limit the access of metaverse-based tools to a minority of patients.

## 8. Cybersecurity for Telemedicine

An ideal telemedicine program with videoconferencing should balance security aspects with user-friendliness for patients and providers. Cost, browser integration, mobile platforms, and electronic health record combinations need also be acknowledged. Although movement disorder specialists have traditionally presented videos of patients at professional meetings, physicians need expert guidance when selecting a videoconferencing software in terms of the legal framework, technical capabilities, licenses, patients’ access, and costs. Compliance with data protection requirements varies worldwide. Examples of data protection regulations include the European Union General Data Protection Regulation (GDPR), a key document intended for protecting personal data in Europe. In the United States, physicians can use the Health Insurance Portability and Accountability Act (HIPAA) compliant software. 

Many videoconferencing software platforms are currently available on the market. We recently reviewed the main technical and security characteristics of commercially available videoconferencing software for healthcare use [69]. Out of twenty-six videoconferencing software solutions, thirteen (50%) were specifically designed for healthcare and six (23%) were compliant with European and US regulations [69]. Overall technical and security information were found in five (19.2%) platforms, including Microsoft Teams, Google Hangout, Coviu, Doxy.me, and Thera. However, itemized information about the technical features and data security of these videoconferencing platforms was not directly retrievable. 

## 9. Barriers and Limitations for Telemedicine

Among the most recognized advantages of telemedicine, several studies have reported increased equity and access to specialized care for patients, greater comfort, and reduced travel time and costs, especially for homebound patients with severe mobility problems [70,71,72]. However, some neurologists are skeptical about the suitability of remote examination intended for new diagnoses, especially in complex cases, given the inherent limitation posed by telemedicine, and thus prefer to conduct traditional follow-up visits [4,73].

Shalash et al. [4] recently reported varying levels of patient comfort and acceptance of telemedicine among different socioeconomic and cultural groups. Specific concerns about telemedicine include the appropriateness of communicating sensitive information, the potential loss of privacy, and some technical barriers regarding computer literacy and poor audio/video quality due to internet connectivity problems. Common inconveniences mentioned by both patients and clinicians included technical problems, lack of hands-on examinations, and reduced quality of doctor–patient communication [74]. In general, doctors find it difficult to communicate bad news to patients over the telephone or during video visits [74]. At present, telemedicine cannot replace in-person healthcare delivery but can be seen as a promising tool suitable for specific settings [6].

## 10. Conclusions

An increasing body of evidence presents telemedicine as a feasible and effective option for the remote management of PD and other movement disorders. Telemedicine is a supportive tool to be considered for selected patient subpopulations. Future studies will confirm the potential of telemedicine to provide accessible and equitable care for patients with neurological conditions.

## Figures and Tables

**Table 1 brainsci-12-01735-t001:** Achievements and challenges of telemedicine.

		Advantages	Disadvantages
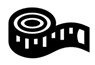	Outcomes-Feasibility-Satisfaction-Efficacy-Reliability-Patient perception	-Necessary for clinical comparison-Easy to analyze	Lack of information on other outcomes, such as cost-effectiveness
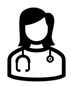	Clinical Care-Tele-expertise-Teleconsultation	-Improve patient’s access to movement disorders specialists-Improve the distribution of qualified health providers-Decrease travel burden	-Limited neurological exam-Limited information on the accuracy of first movement disorder consultation-Reduced quality of doctor–patient contact-Concerns for sensitive information-Limited access to technology in specific settings -Computer literacy-Limited clinician confidence
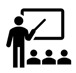	Tele-education-For health professionals-Patients and caregivers	-Prevent academic isolation-Facilitate distance learning-Indirectly improved access to healthcare	-Lack of specific outcomes for tele-education -Limited access to technology in specific settings-Internet connectivity issues-Computer literacy
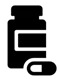	Teletreatment	-Particularly indicated for homebound patients with comorbid diseases-Available for advanced therapies in PD	-Still little safety information-Internet connectivity issues-Computer literacy
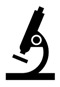	Teleresearch (clinical trials)	-Improve access to research for minorities and underserved areas-Facilitate the collection of biological and safety data	-Limited information -Internet connectivity issues
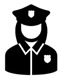	Cybersecurity	-A legal framework has been developed in certain regions of the world.	-Detailed information about technical capabilities and data security of videoconferencing tools are not easily and openly retrievable

## Data Availability

Data availability is not applicable for this article.

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
