# Peer review of "Telemedicine in the Management of Parkinson’s Disease: Achievements, Challenges, and Future Perspectives"

_brainsci, 2022, doi:10.3390/brainsci12121735_

Round 1
Reviewer 1 Report
The main purpose of the work was to achieve greater insight into the telemedicine, with a particular focus on Parkinson's disease
The paper is generally well written however I can't define what kind of paper it is: I think it's an editorial or a narrative review (however, the search criteria of the articles should be added).
I suggest structuring the article as a narrative review by entering the search criteria and the number of selected articles.
· The introduction section includes valid background information. I propose to expand the information on wereable sensor with more directional literature also on telerehabilitation.
Author Response
Thank you for your comment. Regarding the type of paper, we believe that this paper should be better classified as a “perspective” and not as an review with a standard literature review.
Regarding the second comment, we appreciate it but we would like to still focus on the use of telemedicine using a global approach, of course including telerehabilitation (references: 6-9, 59,60,70). We stated that the review of the use of wearable technology is outside the scope of this article, as it would be by itself a different article.
Reviewer 2 Report
Dear Author,
The type of article should be clarified, is it a review article? If yes, what type of review is it?
The methods should be described (how the articles were chosen, what databases were used, inclusion and exclusion criteria, furthermore what were the purposes of the research, etc).
Images or figures should be present.
Also, the title is naming PD, but in the main text is not enough information about PD and telemedicine.
It would be insightful to add some examples of the use of telemedicine in PD, maybe applying to the anamnesis, evolution, treatment, other comorbidity.
Furthermore, it would be useful to describe alternative therapies in PD using telemedicine such as psychotherapy, dance lessons, melotherapy etc.
More information is also needed regarding the achievements, advantages, disadvantages and challenges.
For a review, it lacks complexity and is not long enough.
Also, more bibliography would be needed in order to sustain the theme.
Author Response
We appreciate the reviewer´s comments. As we said for the first reviewer, we believe that this article should be included as a perspective based on the personal view of the authors. For this reason, this article is not structured as a standard review article. Regarding the title, the majority of the examples for telemedicine provided in this article are for PD compared to other movement disorders. We would like to be inclusive for other movement disorders, because after searching for telemedicine in the literature, approximately 90% of the articles are for PD. We need to report other uses, otherwise, the literature is always biased for PD.
We have provided the best examples for interdisciplinary approach for movement disorders including articles with evidence class I or II. The main problem with non-pharmacological telemedicine interventions are: 1) the scarcity of good quality publications; 2) the extrapolation of the current data for PD and other movement disorders.
Regarding the last point, we also appreciate this comment, but advantages and disadvantages have been extensively described by other authors and we would not like to be too repetitive. The originality of this article is to combine different pieces of telemedicine including cybersecurity to provide this global perspective.
Round 2
